# Ethanol Induces Extracellular Vesicle Secretion by Altering Lipid Metabolism through the Mitochondria-Associated ER Membranes and Sphingomyelinases

**DOI:** 10.3390/ijms22168438

**Published:** 2021-08-05

**Authors:** Francesc Ibáñez, Jorge Montesinos, Estela Area-Gomez, Consuelo Guerri, María Pascual

**Affiliations:** 1Department of Molecular and Cellular Pathology of Alcohol, Príncipe Felipe Research Center, 46012 Valencia, Spain; fibanez@cipf.es (F.I.); cguerri@cipf.es (C.G.); 2Department of Neurology, Columbia University Medical Center, New York, NY 10032, USA; eag2118@cumc.columbia.edu; 3Department of Physiology, School of Medicine and Dentistry, University of Valencia, 46010 Valencia, Spain

**Keywords:** extracellular vesicles, lipid metabolism, mitochondria-associated ER membranes, alcohol, neuroinflammation, microglia, sphingomyelinases, phospholipids

## Abstract

Recent evidence pinpoints extracellular vesicles (EVs) as key players in intercellular communication. Given the importance of cholesterol and sphingomyelin in EV biology, and the relevance of mitochondria-associated endoplasmic reticulum membranes (MAMs) in cholesterol/sphingomyelin homeostasis, we evaluated if MAMs and sphingomyelinases (SMases) could participate in ethanol-induced EV release. EVs were isolated from the extracellular medium of BV2 microglia treated or not with ethanol (50 and 100 mM). Radioactive metabolic tracers combined with thin layer chromatography were used as quantitative methods to assay phospholipid transfer, SMase activity and cholesterol uptake/esterification. Inhibitors of SMase (desipramine and GW4869) and MAM (cyclosporin A) activities were also utilized. Our data show that ethanol increases the secretion and inflammatory molecule concentration of EVs. Ethanol also upregulates MAM activity and alters lipid metabolism by increasing cholesterol uptake, cholesterol esterification and SMase activity in microglia. Notably, the inhibition of either SMase or MAM activity prevented the ethanol-induced increase in EV secretion. Collectively, these results strongly support a lipid-driven mechanism, specifically via SMases and MAM, to explain the effect of ethanol on EV secretion in glial cells.

## 1. Introduction

Extracellular vesicles (EVs), known as exosomes, are bilayered lipid membrane particles secreted from cells that play an important role in intercellular communication under physiological and pathological conditions. These nanovesicles (30–150 nm in diameter) carry specific proteins, RNAs and lipids [1]. The internalization of EVs secreted by bystander cells could modulate signaling pathways, lead to long-term changes in cellular behavior and play important roles in human diseases, such as inflammation and neurodegenerative disorders [2].

Studies have shown that microglia increase EV secretion after a stimulus [3,4], and the content of microglia-derived EVs differs depending on the nature of the stimuli, triggering either pro- or anti-inflammatory responses [5,6,7]. Interestingly, drugs of addiction such as cocaine [8] or ethanol [9] have also been shown to alter microglial EV secretion. Specifically, ethanol was reported to increase the content of pro-inflammatory cytokines and chemokines in microglia-derived EVs in a dose-dependent manner [9].

In agreement, using cultured astroglia cells, we have shown that ethanol treatment increases the release of EVs and enriches their content in inflammatory-related proteins and miRNAs, effects that are associated with Toll-like receptor 4 (TLR4) immune response activation [10]. Our previous studies have demonstrated that ethanol, by interacting with cholesterol/sphingomyelin-enriched lipid rafts, can trigger a TLR4 response as well as induce different signaling pathways, such as MAPKs or NFκB [11,12]. TLR4 signaling activation leads to the release of cytokines and inflammatory mediators, and causes neural damage [13]. However, the ethanol-induced inflammatory response [11,12] and the upregulation of proinflammatory EV secretion [10] are abolished in glial cells from TLR4-deficient mice.

The lipid metabolism has been extensively implicated in EV secretion dynamics [14,15], but the connection between lipids and EV secretion upon ethanol stimulation remains unexplored. However, it is known that EVs are enriched in both cholesterol and sphingomyelin, which are the main lipid components of lipid rafts [16]. In fact, the hydrolysis of sphingomyelin into ceramide by sphingomyelinases (SMases) has been closely associated with the biogenesis of EVs [17]. In addition, direct modification of the plasma membrane content of either cholesterol/sphingomyelin [18] or phosphatidylserine [19] promotes microvesicle formation. Interestingly, mitochondria-associated endoplasmic reticulum (ER) membranes (MAMs), a specialized subdomain of the ER with features of lipid rafts, have been recently linked with cholesterol/sphingomyelin and phospholipid homeostasis in the cell, among other important metabolic processes [20,21].

Based on our previous results demonstrating that ethanol increases the number of secreted EVs in astroglial cells, we hypothesized that ethanol would promote EV formation and secretion via the upregulation of MAM-localized activities. In this study, we demonstrate that ethanol up-regulates MAM-localized phospholipid transfer and cholesterol esterification. Strikingly, the increase in EV secretion induced by ethanol decreases in the presence of MAM and SMase inhibitors, which suggests a link between EV secretion and MAM and SMase activities.

## 2. Results

### 2.1. Ethanol-Induced Proinflammatory BV2 Microglia Increase EV Release and the Concentration of Proinflammatory Molecules in EVs

We have demonstrated that ethanol is able to activate glial cells in primary cultures, which leads to an inflammatory immune response [11,12]. Herein we wondered whether physiologically relevant concentrations of ethanol can activate the BV2 microglial cell line. Figure 1A shows that ethanol increased the phosphorylation of ERK and p38 after 30 min of treatment, although maximal activation was noted between 1 and 3 h and decreased thereafter. Moreover, the phosphorylation of the NFκB subunit p65 was found to increase upon 3 h of stimulation. As microglia activation is associated with increased phagocytic activity, we also measured the phagocytic capacity of ethanol-treated microglial cells. To do so, BV2 cells treated with or without ethanol were exposed to fluorescein-labeled latex beads for 30 min (Figure 1B). Ethanol-treated cells exhibited a more robust phagocytic response after 24 h of treatment than untreated cells, which indicates that ethanol can induce a phagocytic response in BV2 cells. The internalization of fluorescein-labeled latex beads in the cytoplasm was confirmed using the xyz axes projections obtained using confocal microscopy (Figure 1B).

Our recent studies have shown that ethanol up-regulates astrocyte-derived EV release and increases their concentration of neuroinflammatory proteins and miRNAs [10]. We, therefore, analyzed the release and content of EVs from the supernatant of the ethanol-treated and untreated BV2 cells. By means of transmission electron microscopy, we observed round-shaped particles ranging from 100 to 200 nm in size according to the exosome size (Figure 1C). These data were corroborated by a nanoparticle tracking analysis, in which the highest peak of the secreted nanoparticles ranged between 100 and 200 nm (Figure 1D). Furthermore, ethanol significantly increased both the number of particles detected between 100 and 200 nm (Figure 1D) and the expression of exosomal markers HSP70 and CD81 (Figure 1E) compared to the untreated cells.

Ethanol treatment increased not only the levels of inflammatory proteins, such as TLR4, NLRP3 and IL-1R in the EVs that derived from BV2 cells (Figure 1F), but also the mir-146a-5p, mir-21-5p and let-7b levels significantly versus their control counterparts (Figure 1G). These microRNAs have been associated with the activation of the TLR4/NF-κB signaling pathway [10,22,23].

### 2.2. Ethanol Upregulates MAM-Mediated Phospholipid Transfer Activity in Brain-Derived Crude Membranes and in BV2 Microglial Cells

Physiologically low ethanol doses have been found to trigger neuroinflammation by recruiting TLR4 into plasma membrane lipid rafts, while high ethanol doses suppress TLR4 signaling via the disruption of lipid raft clustering [24]. We, therefore, hypothesized that MAM, as a lipid raft ER subdomain, would be sensitive to the membrane-fluidifying effect of ethanol. To test this hypothesis, we employed an established phospholipid transfer (PLT) assay as a proxy of MAM activity [25] to track the incorporation of radioactive serine into phosphatidylserine (PtdSer), and its conversion into phosphatidylethanolamine (PtdEtn) and phosphatidylcholine (PtdCho).

As a first approach, we assessed the MAM response to ethanol in the crude membrane preparations isolated from murine brains, as detailed in the Materials and Methods section. Figure 2A shows that ethanol significantly increased the radioactive PtdSer levels in crude membrane fractions compared to the controls. This effect occurred in a dose-dependent manner. However, the radioactive PtdEtn levels showed similar alterations, but only at the 50 or 100 mM doses. Finally, radioactive PtdCho displayed no significant alterations, but a slight increase, mainly at 50 mM. Based on the results obtained in Figure 2A, we decided to corroborate the effect of ethanol on MAM in a cellular model using the BV2 cell line and two ethanol treatment paradigms. First, an acute cotreatment with ^3^H-serine and ethanol, either at 50 or 100 mM (10 mM had only a slight effect on MAM activity; data not shown), was employed (Figure 2B). In parallel, we also pre-treated BV2 cells for 24 h with the 50 or 100 mM ethanol dose. Then, ^3^H-serine was added to track MAM activity (sustained treatment; Figure 2C). In either paradigm, ethanol was able to induce an enhanced MAM activity in a dose-dependent manner. Interestingly, the 100 mM ethanol dose of the acute treatment had the strongest impact on MAM activity, and increased by ~2–3-fold compared to the untreated control. It is noteworthy that MAM activity remained significantly up-regulated at 24 h of ethanol treatment, albeit at a lower level than those found in the acute treatment.

### 2.3. Ethanol Upregulates Cholesterol Metabolism and SMase Activity in BV2 Cells

Previous studies have identified the sustained trafficking of cholesterol to MAM as a cause for MAM up-regulation in neurodegenerative disease [26]. Therefore, we sought to analyze whether ethanol was able to increase cholesterol uptake and its concomitant translocation to MAM.

In order to answer these questions, the uptake of radioactive cholesterol and its conversion into cholesteryl esters via MAM-resident enzyme, ACAT1 (Acyl-CoA: cholesterol acyltransferase), were analyzed [27]. Figure 3A shows that BV2 cells incubated with 50 mM ethanol significantly increased cholesterol uptake compared to the untreated control. The levels of ^3^H-cholesterol were also upregulated by 100 mM ethanol, but they failed to reach statistical significance.

The results obtained when analyzing cholesterol esterification showed that 50 mM of ethanol was able to up-regulate the incorporation of cholesterol into cholesteryl esters in BV2 cells versus untreated cells. Strikingly, a 100 mM dose of ethanol did not alter the cholesteryl ester levels (Figure 3B).

It is known that, in lipid rafts, the interaction of cholesterol with sphingomyelin acts as shielding from the aqueous phase [16]. Thus, the hydrolysis of the “shielding” sphingomyelin into ceramide by the action of neutral or acidic SMase can facilitate the mobilization of lipid raft-bound cholesterol [28]. Our previous studies have demonstrated that ethanol can trigger the generation and accumulation of SMase-derived ceramide in astrocytes in culture [29]. In order to corroborate these results in the BV2 cell line, we analyzed the effect of ethanol on SMase activity by the conversion of water-insoluble ^3^H-sphingomyelin into water-soluble ^3^H-phosphocholine. Figure 3C shows that both the 50 and 100 mM doses significantly increased SMase activity compared to the untreated control.

### 2.4. Ethanol-Induced Exosome Secretion Is Reversed by Inhibition of Either MAM or SMases

Our previous results suggest that ethanol could modulate cholesterol uptake and SMase activation, and both events have been previously associated with the up-regulation of MAM activity [26,30]. Therefore, to address whether SMase activity is involved in the ethanol-induced up-regulation of MAM activity, we used inhibitors for the neutral and acidic SMases, such as desipramine and GW4869, respectively. We also used cyclosporin A as an indirect disruptor of MAM activity due to its ability to block the interaction between Cyclophilin D and IP3R, and concomitantly, the connection between mitochondria and the ER at the MAM interface [31]. Figure 4A depicts how ethanol at 50 mM and 100 mM was able to up-regulate MAM activity by increasing ^3^H-serine incorporation into PtdSer, PtdEtn and PtdCho. Interestingly, these effects were abolished when the cells were treated with ethanol along with GW4869 or cyclosporin A, which implies the involvement of SMase activity in ethanol-induced MAM up-regulation. However, no significant changes were observed when desipramine was used to inhibit ethanol-induced MAM activity.

SMase activity is a pivotal step in exosome generation [14,32]. In fact, GW4869 is a widely used inhibitor of exosome biogenesis for its ability to block the ceramide-mediated inward budding of multivesicular bodies and to prevent the release of exosomes [33]. As we have demonstrated that ethanol up-regulates SMases and MAM activities, we sought to analyze their involvement in the increased EV secretion upon ethanol treatment. We compared the secreted EV levels using Nanosight NS300 upon the inhibition of acidic SMase, neutral SMase or MAM using desipramine, GW4869 or cyclosporin A, respectively. Figure 4B corroborates that ethanol is indeed able to induce greater EV secretion in BV2 cells compared to the untreated conditions. Interestingly, when either of the inhibitors was administered along with ethanol, the number of secreted EVs lowered to control levels. These results confirm the role of SMase activity in exosome release and suggest the relevance of MAM in this process.

## 3. Discussion

Our previous studies have demonstrated that ethanol, through the activation of TLR4 in glial cells, is able to induce a neuroinflammatory response that leads to the release of cytokines and inflammatory mediators, and causes brain damage and neurodegeneration [11,12]. Recent evidence has further shown that ethanol increases EV secretion in astroglial cells, and also up-regulates the concentration of inflammatory proteins and miRNAs in EVs to cause neural cell death [10]. Although the mechanisms underlying ethanol-induced EV secretion remain unknown, we herein demonstrate that ethanol, through the activation of MAM and SMases, induces greater EV secretion in microglial BV2 cells as the disruption of MAM or the inhibition of SMase abolishes ethanol-induced EV release. These results also support the role of the cholesterol metabolism in this process as ethanol induces increased free cholesterol uptake and its incorporation into cholesteryl esters.

The well-established effects of ethanol on plasma membrane fluidity have been associated with changes in membrane properties and the alteration of cellular functions [34]. In fact, our previous results have demonstrated that low ethanol concentrations are able to modify lipid raft fluidity, which would facilitate TLR4 activation [24]. Accordingly, the results of this study revealed the susceptibility of MAM to the membrane fluidifying effects of ethanol because phospholipid transfer activity was affected by ethanol in a dose-dependent manner. Thus, we can suggest that the increased MAM activity induced by ethanol when crude membranes were treated with different ethanol doses could be due to enhanced membrane fluidity. Although specific assays to monitor membrane fluidity under these conditions would further clarify this, with a BV2 cellular model we were able to recapitulate those findings to support the notion that ethanol upregulates MAM-localized activities.

Cholesterol and sphingomyelin are pivotal components for maintaining lipid raft structures in cellular membranes. Indeed, cholesterol and sphingomyelin homeostasis and metabolism are tightly co-regulated [28]. Cellular cholesterol levels are regulated through the crosstalk between the plasma membrane, where most cellular cholesterol resides, and the ER, where the protein machinery that regulates cholesterol levels is found [35]. Interestingly, the internalization of cholesterol and its transport to the ER have been shown to induce the formation of MAM domains [26], along with the activation of SMases to facilitate membrane-bound cholesterol translocation.

Our results indicate that ethanol up-regulates cholesterol uptake and SMase activity in BV2 cells. An increased cholesterol uptake has been demonstrated to facilitate LPS-induced microglial activation, along with enhanced phagocytosis [36]. It has been suggested that the depletion of membrane-bound cholesterol could increase membrane fluidity and, in turn, facilitate phagocytosis [37]. Interestingly, high cholesterol levels are associated with increased neuroinflammation and cognitive deficits [38]. Indeed, the manipulation of cholesterol levels in microglia can modify both cytokine release and phagocytosis [39]. Some reports have revealed that the cholesterol loading of the plasma membrane induces TLR4-dependent signaling, while TLR3 signaling is also engaged when endosomal compartments are loaded with cholesterol [40]. This scenario suggests that specific lipid alterations may depend on different stimuli/activation states. Likewise, ethanol can trigger different signaling pathways depending on the dose, timing and subcellular compartment. Constant cholesterol transport between the plasma membrane and the ER is pivotal for regulating the total cholesterol content [41], and when cholesterol is present in the ER, it is esterified to maintain membrane homeostasis. We reveal that ethanol treatment at 50 mM increased cholesterol esterification. Indeed, microglia accumulating cholesteryl esters showed a reactive state [42] that has been associated with neurodegenerative phenotypes [43]. However, aged proinflammatory microglia have been found to accumulate lipid droplets composed of triacylglycerides instead of cholesteryl esters, while showing defective phagocytosis [44].

Sphingomyelin hydrolysis into ceramide is catalyzed by SMases. Different types of SMases have been classified according to the pH at which they exhibit maximal activity: acid, neutral or alkaline. Although these enzymes are located in the plasma membrane, recent evidence supports the notion that the enzymatic activity of other subcellular regions, such as Golgi or MAM, is enriched [30]. The neutral SMase is involved in many cellular responses, such as cell growth arrest, exosome formation and the inflammatory response [45,46], and is one of the first responders to cellular stress [47]. Here, we show that ethanol is able to increase neutral SMase activity in microglia, which could be associated with MAM activation and EV release. It is known that neutral SMase participates in IL-6 release in mouse neuronal cultures through the activation of the neutral SMase/Src kinase signaling pathway, effects that can be associated with acute brain injury [48]. The suppression of ceramide generation via acidic SMase inhibition or de novo ceramide synthesis decreases LPS-induced cytokine secretion in Hep3B cells, which might be involved in hepatic inflammatory processes [49]. Pascual et al. [29] have shown that ethanol treatment is able to induce SMase activity in astroglial cells, which leads to their increased ceramide content and cell death.

EVs’ biogenesis and release are complex processes that involve many proteins and subcellular systems [1]. Two different pathways for EV biogenesis have been described, the ESCRT dependent pathway driven by RAS, and the ESCRT-independent pathway driven by neutral SMase [14]. Recent studies have found that the inhibition of neutral SMase blocks ceramide production, which is needed for the inward budding and shedding of intraluminal vesicles. Thus, exosome biogenesis and release diminish. However, acidic SMase inhibition affects the release of microvesicles, which are bigger particles than exosomes [32]. Here, we show that ethanol activates neutral SMase and increases EV secretion, which are reverted when ethanol is used along with SMase inhibitors (neutral or acid SMase), in agreement with recent reports using primary cultured microglia [9]. What this suggests is that ethanol promotes EV secretion by activating the ESCRT-independent pathway. It is noteworthy that neutral SMase inhibition suppresses ethanol-induced EV secretion and MAM activity, while acidic SMase inhibition blocks ethanol-induced EV secretion and has no effect on MAM-localized PLT activity. Our results further demonstrate that MAM inhibition leads to the same effect that is observed when inhibiting SMases. These results support a connection between MAM and neutral SMase activity. Indeed, recent reports have described that cellular models of MAM up-regulation promote neutral SMase recruitment to MAM, which results in higher ceramide levels [30].

MAM may also be implicated in EV biogenesis and release via lipid regulation. Some studies have shown that the balance between PtdSer and PtdEtn could promote vesicle budding [50], and the cholesterol metabolism can influence EV biogenesis [51]. These processes are regulated by MAM. On the effect of cholesterol on exosomal release, recent reports have revealed that cholesterol loading triggers exosomal release and the upregulation of inflammatory markers in macrophages [52], while opposite effects have been observed when cholesterol synthesis is pharmacologically inhibited [51]. However, cholesterol reduction has been shown to upregulate exosomal release in PC-3 cells [53]. Elegant studies from [54] have demonstrated that Caveolin-1, the main component of lipid-raft caveolae, regulates exosome biogenesis and sorting via cholesterol redistribution within the cell. Interestingly, caveolin-1 has been previously described as a protective factor in alcohol-induced inflammation and ROS generation [55]. These results support the idea that MAM, by up-regulating cholesterol and phospholipids (which are building blocks for the formation of EVs) and activating SMases, could mediate EV biogenesis and release.

These findings also suggest that the use of SMase inhibitors could be good candidates as therapeutic targets to diminish EV secretion in microglial cells to avoid the amplification of the ethanol-induced neuroinflammation. Recently, in vivo models of postnatal alcohol exposure showed an induction of the release of exosomes from microglia in hypothalamic tissue, along with neurotoxicity through caspase activation and ROS formation [9]. Additionally, interestingly, treatment with a SMase inhibitor successfully prevented neurotoxic damage.

It is important to point out the limitations of our results. For instance, the use of a cell line has an important disadvantage as these cells might not have the relevant attributes or functions of primary cell cultures or in vivo animal models. Another limitation is the possible off-targets effects of the SMase and MAM inhibitors employed herein; therefore, further studies using the genetic manipulation of SMase or MAM activities could help to corroborate the proposed mechanism of ethanol in the release of EVs.

Taken together, these results support the idea that MAM, by upregulating cholesterol and phospholipids (which are building blocks for the formation of EVs) and activating SMases, could mediate EV biogenesis and release.

## 4. Materials and Methods

### 4.1. Animals

C57/BL6 wild-type (WT) male mice (The Jackson Laboratory, Bar Harbor, ME, USA) were used (*n* = 5). Mice were distributed into 3–4 animals per cage and maintained with water and solid diet ad libitum under controlled conditions of temperature (23 °C), humidity (60%) and light/dark cycles (12 h/12 h). All the experiments were performed according to a protocol (AC-AABG5550, approved date 6 May 2020) approved by the Institutional Animal Care and Use Committee of the Columbia University Medical Center, and were consistent with the National Institutes of Health Guide for the Care and Use of Laboratory Animals.

### 4.2. Cell Cultures

For this study, we used the immortalized mouse microglial cell line, BV2 (Innoprot S.L., Bizkaia, Spain). BV2 cells were cultured using Dulbecco’s modified Eagle serum (DMEM) supplemented with 10% fetal bovine serum (FBS), 100 U/mL of penicillin/streptomycin and 2.5 µg/mL of fungizone, and were seeded at 850 cells/mm^2^.

### 4.3. Ethanol Treatment

For the time-course studies, FBS was replaced with Bovine Serum Albumin (free fatty acid BSA, 1 mg/mL) 2 h prior to ethanol (50 mM) stimulation, then BV2 cells were harvested at the indicated times (0, 30 min, 1, 3, 7 and 24 h), frozen and stored at −80 °C until further use.

### 4.4. Phagocytosis

BV2 microglial cells were cultured on glass coverslips and incubated with 2-micrometer-diameter FluoSpheres at the 0.01% solid mass (Molecular Probes, Eugene, OR, USA), as previously described [12]. To verify the uptake by phagocytosis, Z-scan images through microglia cells were obtained using a Leica confocal microscope to corroborate that the fluorescein-labeled latex beads had been internalized and were not above the cell or had adhered to the cell surface. Appendix A shows the panoramic images and fluorescence composition of Figure 1B.

### 4.5. Exosome Isolation by Ultracentrifugation

For the EV studies, BV2 cells were rinsed and incubated in serum-free medium with BSA for 2 h. The cells were then treated with ethanol (50 mM) for 24 h. In some cases, a pretreatment with the inhibitors desipramine (5 µM), GW4869 (10 µM) or cyclosporin A (2 µM) was carried out 2 h before initiating ethanol treatment. Then, media were collected and centrifuged at 300× *g* and 4 °C for 10 min. Supernatants were recovered and centrifuged at 2000× *g* and 4 °C for 20 min. Once again, supernatants were collected and ultracentrifuged for 30 min at 4 °C and 10,000× *g*. The resultant supernatants were collected, transferred to fresh tubes and spun at 100,000× *g* and 4 °C for 1 h. Immediately afterward, supernatants were discarded, and pellets were resuspended in PBS and spun again under the same conditions. The pellets containing an exosome-rich fraction were resuspended in the PBS-containing protease inhibitors. Protein levels were determined using the BCA assay and the amount of mg protein/mL in the different experimental conditions was adjusted to ~10 mg/mL.

### 4.6. Exosome Characterization by Transmission Electron Microscopy

Freshly isolated EVs were fixed with 2% paraformaldehyde and were prepared as previously described [10]. Preparations were examined under a transmission FEI Tecnai G2 Spirit electron microscope (FEI Europe, Eindhoven, The Netherlands) with a digital camera Morada (Olympus Soft Image Solutions GmbH, Münster, Germany). Appendix A shows the panoramic images of Figure 1C.

### 4.7. Nanoparticles Tracking Analysis

An analysis of the absolute size range and concentration of microvesicles was performed using NanoSight NS300 Malvern (NanoSight Ltd., Minton Park, UK), as previously described [10]. Appendix A shows the size range distribution of the EVs isolated from the supernatant of BV2 cells.

### 4.8. Western Blot Analysis

Equal amounts of protein were loaded and separated in SDS-PAGE gels and transferred to PVDF membranes as previously described [10]. The employed antibodies were anti-HSP70, anti-CD81, anti-ERK, anti-p-ERK, anti-p38 and anti-p-p38 (Santa Cruz Biotechnology, Dallas, TX, USA). Membranes were incubated with the respective anti-HRP secondary antibodies and developed using the ECL system (ECL Plus; ThermoFisher Scientific, Hanover Park, IL, USA). Band intensity was quantified with the ImageJ 1.44p analysis software. The densitometric analysis is shown in arbitrary units and normalized to its dephosphorylated form. Appendix A includes the whole membrane of each protein expression.

### 4.9. RNA Isolation, Reverse Transcription and Quantitative RT-PCR

The total RNA of EVs was isolated following the manufacturer’s instructions (Total Exosome RNA Isolation Kit, Invitrogen, Lithuania). Total miRNAs were reverse transcribed using the TaqMan Advanced miRNA Assays (ThermoFisher Scientific, Hanover Park, IL, USA), respectively. Quantitative Two-Step RT-PCR (real-time reverse transcription) was performed with the Light-Cycler 480 detection System (Roche Diagnostics, Basel, Switzerland). Specific miRNAs assays were amplified using the TaqMan Fast Advanced Master Mix (ThermoFisher Scientific, Hanover Park, IL, USA). Data were analyzed using the LightCycler 480 relative quantification software. The nucleotide sequences of the used miRNAs assays are detailed in the Appendix A.

### 4.10. Cholesterol Trafficking and Esterification Assays

Cholesterol trafficking and esterification were measured as previously described [30]. Cells were incubated for 2 h in serum-free medium to ensure the removal of exogenous lipids. Next, 2.5 μCi/mL of ^3^H-cholesterol (complexed with 2% fatty acid free-BSA), along with 50 or 100 mM of ethanol, were added to cells for the indicated time periods. Cells were then washed, scraped and collected in PBS. Equal protein amounts were used to extract lipids using three chloroform:methanol volumes (2:1 *v*/*v*). After vortexing and centrifugation at 8000 *g* for 5 min, the organic phase was recovered and blown to dryness under nitrogen. Dried lipids were resuspended in 30 µL of chloroform:methanol (2:1 *v*/*v*) and applied to a TLC plate. A hexanes/diethyl ether/acetic acid (80:20:1 *v*/*v*/*v*) mixture was used as the solvent. The iodine-stained bands corresponding to free cholesterol and cholesteryl esters (identified by co-migrating standards) were scraped and counted in a scintillation counter (Packard Tri-Carb 2900TR, GMI, Ramsey, MN, USA).

### 4.11. Sphingomyelinase Activity

SMase activity was assayed as previously described [30]. A total of 100 µg of protein were incubated with 400 µM of bovine brain sphingomyelin spiked with 22,000 dpm of [^3^H]-bovine sphingomyelin (1 nCi/sample) in 100 mM Tris/glycine, pH 7, 1.55 mM of Triton X-100, 0.025% BSA and 1 mM of MgCl_2_. Reactions were carried out in borosilicate glass culture tubes at 37 °C overnight, followed by quenching with 1.2 mL of -cold 10% trichloroacetic acid, incubation at 4 °C for 30 min and centrifugation at 380× *g* at 4 °C for 20 min. Then, 1 mL of supernatant was transferred to clean tubes, 1 mL of ether was added, and the mixture was vortexed and centrifuged at 380× *g* for 5 min. Finally, 800 µL of the aqueous bottom phase (containing the generated ^3^H-phosphocholine) was transferred to scintillation vials with 5 mL of Scintiverse BD (ThermoFisher Scientific, Hanover Park, IL, USA) and measured in a Scintillation Counter (Packard Tri-Carb 2900TR, GMI, Ramsey, MN, USA).

### 4.12. Analysis of Phospholipid Synthesis in Subcellular Fractions

Crude membrane fractions were freshly isolated from C57/BL6 adult mice brain as described elsewhere [56]. Two hundred micrograms of crude membrane preparation were incubated in a final volume of 200 μL of phospholipid synthesis buffer (10 mM of CaCl_2_, 25 mM of HEPES, pH 7.4, and 3 μCi/mL ^3^H-Ser) at different ethanol concentrations (0, 10, 50, 100 mM) for 30 min at 37 °C. The reaction was stopped by adding three volumes of chloroform/methanol (2:1, *v*/*v*). After centrifugation at 8000 *g* for 5 min, the organic phase was washed twice with two volumes of methanol/water (1:1, *v*/*v*), and the organic phase was recovered and blown to dryness under nitrogen. Dried lipids were resuspended in 60 μL of chloroform/methanol 2:1 and applied to a TLC plate. Phospholipids were separated using two solvents composed of petroleum ether/diethyl ether/acetic acid (84:15:1, *v*/*v*) and chloroform/methanol/acetic acid/water (60:50:1:4, *v*/*v*). Development was performed by exposing the plate to an iodine vapor. The spots corresponding to the relevant phospholipids (identified by co-migrating standards) were scraped and counted in a scintillation counter (Packard Tri-Carb 2900TR, GMI, Ramsey, MN, USA) [25].

### 4.13. Analysis of Phospholipid Synthesis in Cultured Cells

To assay the effect of ethanol on phospholipid synthesis in BV2 cells, two ethanol treatment paradigms were used. In one paradigm, ethanol (50 and 100 mM) was added along with 2.5 μCi/mL of ^3^H-Serine, and cells were recollected at 2 and 4 h after both treatments (acute treatment). In the other paradigm, microglial cells were pretreated with ethanol (50 and 100 mM) for 20 h and then incubated with 2.5 μCi/mL of ^3^H-Serine. Cells were recollected 2 and 4 h after ^3^H-Serine treatment (sustained treatment). In some cases, a pretreatment with different inhibitors desipramine (5 µM), GW4869 (10 µM) or cyclosporin A (2 µM) was carried out 2 h before initiating ethanol treatment. Then, cells were washed and collected in DPBS, and pelleted at 2500× *g* for 5 min at 4 °C. The cellular pellet was stored at −80 °C until further use. The pellet was resuspended in 0.2 mL of water by removing a small aliquot for protein quantification. Lipid extraction and TLC analysis were performed as described above.

### 4.14. Statistical Analysis

Data represent mean ± SEM. The statistical analysis was performed using GraphPad Prism v6.01 (GraphPad Software Inc., San Diego, CA, USA). All statistical parameters are reported in Appendix A. Shapiro–Wilk test was used to test for data distribution normality. As indicated per each figure legend, *t*-test or one-way ANOVA were used as parametric tests, and Wilcoxon-Mann–Whitney test or Kruskal–Wallis test were used as non-parametric alternatives. Values of *p* < 0.05 were considered statistically significant.

## 5. Conclusions

Taken together, our results demonstrate that low concentrations of ethanol activate MAM and SMases, alter cholesterol metabolism and increase EV secretion in BV2 microglial cells. Indeed, these results highlight the role of MAM and SMases in ethanol-induced enhanced EV secretion, which opens up new avenues for intervention that may attenuate inflammatory response amplification by diminishing ethanol-induced inflammation-associated EV release.

## Figures and Tables

**Figure 1 ijms-22-08438-f001:**
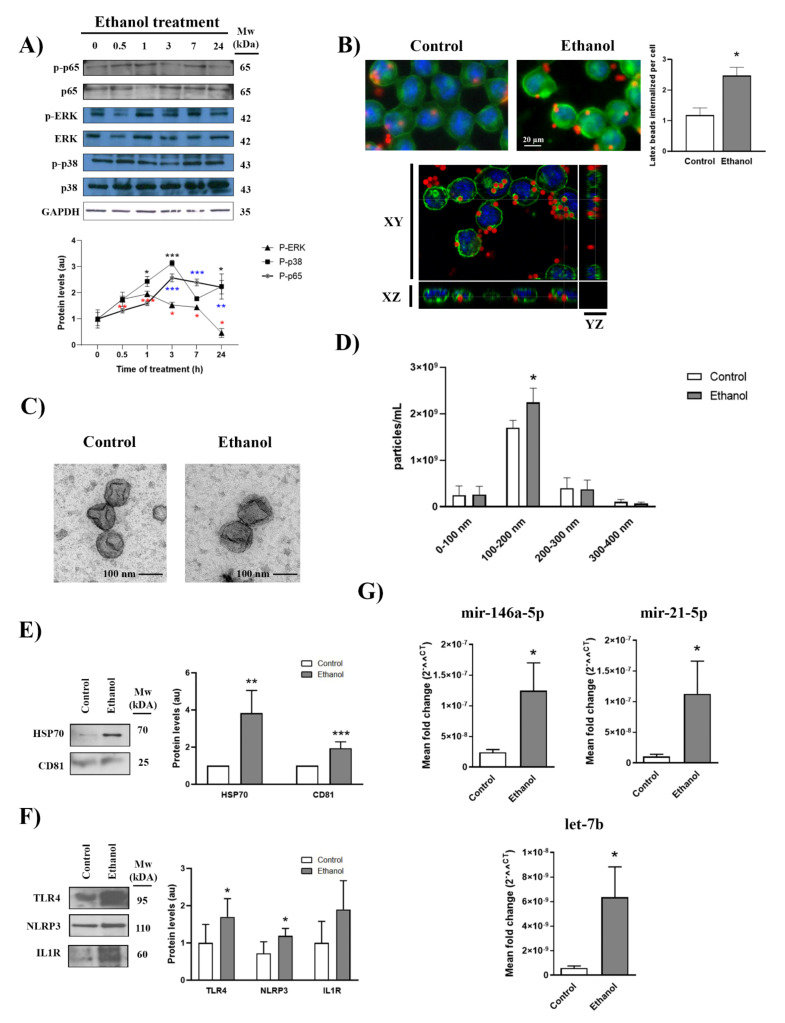
Ethanol activates BV2 microglial cells, increases EV release and alters EV inflammatory molecule concentration. (**A**) Immunoblot analysis and quantification (au, arbitrary units) of p-ERK, p-p38 and p-p65 in cell extracts of ethanol (50 mM)-treated BV2 cells for different time periods (0, 0.5, 1, 3, 7 and 24 h). Blots were stripped, and the total quantities of ERK, p38, p65 and GAPDH were also assessed. A representative immunoblot of each protein is shown. (**B**) The BV2 cells treated with or without ethanol for 24 h were exposed to fluorescein-labeled latex beads for 30 min to internalize them, which reveals the phagocytic activity of these cells. Cells were labeled with tomato lectin. Scale bar, 20 µm. A representative photomicrograph from three different experiments is shown. Graph bars represent the mean number of latex beads phagocytosed by cells for each condition. The xyz axes projections obtained using confocal microscopy (lower panel) show that latex beads are internalized in the cytoplasm. Green channel and blue channel represent cell membranes and nuclei staining, respectively. Appendix A shows the panoramic images and fluorescence composition. (**C**) Electron microscopy images of microglia-derived EVs. Scale bar, 100 nm. Appendix A shows the panoramic images. (**D**) Measurement of the absolute size range and concentration of the EVs derived from microglia by the nanoparticles tracking analysis. (**E**,**F**) Analysis of the levels of the exosome protein markers CD81 and HSP70 (**E**) and the inflammatory-related proteins TLR4, NLRP3 and IL-1R (**F**) present in microglia-derived EVs upon 50 mM ethanol stimulation. A representative immunoblot for each protein and their molecular weight are shown. In each lane, 30 μg of protein were loaded. CD81 was used as the loading control. (**G**) Bar graphs represent the expression of the following miRNAs; miR-146a-5p, miR-21-5p and let-7b; after ethanol treatment (or no treatment) in BV2 microglial cells. Data represent mean ± SEM, *n* = 5 independent experiments. * *p* < 0.05, ** *p* < 0.01 and *** *p* < 0.001 compared to their respective control group according to the *t*-test. For the time-course experiments and nanoparticle tracking analysis, Kruskal–Wallis and One Way ANOVA analysis were used followed by Bonferroni post hoc test, respectively.

**Figure 2 ijms-22-08438-f002:**
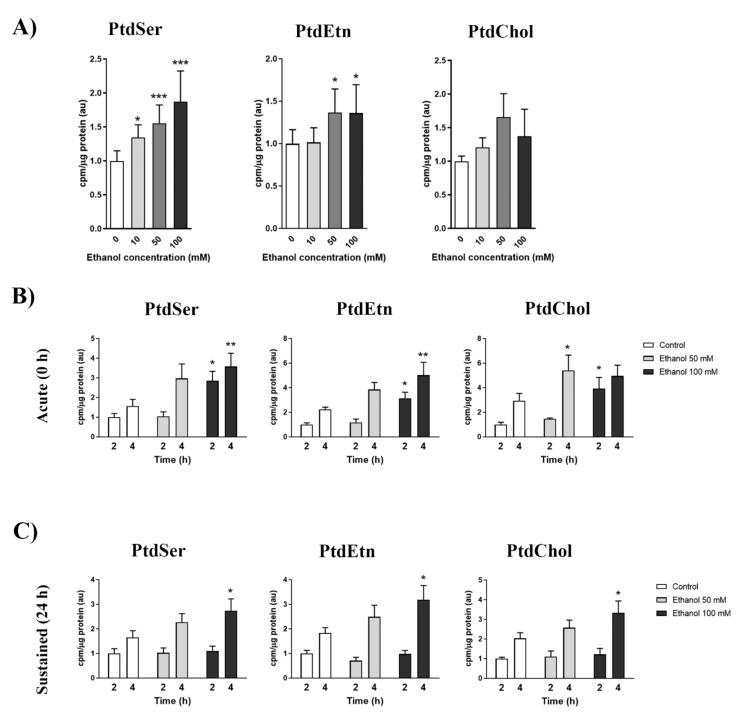
Effects of ethanol on MAM activity in brain-derived crude membranes (**A**) and BV2 cells (**B**,**C**). (**A**) Ethanol effect (0, 10 mM, 50 mM and 100 mM) on the incorporation of ^3^H-Serine into phosphatidylserine (PtdSer), phosphatidylethanolamine (PtdEtn) or phosphatidylcholine (PtdCho) in crude membrane fractions isolated from murine brains. Reaction was carried out for 30 min at 37 °C. Graph bars represent fold change over the control. (**B**,**C**) Time-course analysis of phospholipid synthesis activity in microglial cells after different ethanol treatments. (**B**) The serine-starved BV2 cells were cotreated with ^3^H-Serine along with ethanol (50 or 100 mM) for the indicated times. (**C**) After a 24-hour treatment with ethanol (50 or 100 mM), BV2 cells were serine-starved for 1 h prior to incubation with ^3^H-Serine for the indicated time points. Graph bars represent fold change over the control. Data represent mean ± SEM, *n* = 5 independent experiments. * *p* < 0.05, ** *p* < 0.01 and *** *p* < 0.001 compared to their respective control group according to the one-way ANOVA analysis. Kruskal–Wallis test was used in case of non-Gaussian distribution data. For the PLT in BV2, a two-way ANOVA with Bonferroni as the post hoc test was carried out.

**Figure 3 ijms-22-08438-f003:**
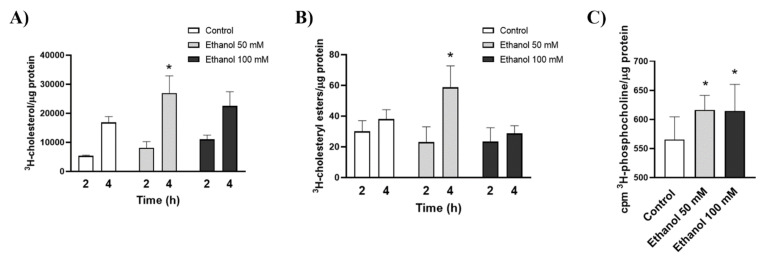
Ethanol treatment alters cholesterol trafficking, esterification and SMase activity. (**A**) Time-course analysis of ^3^H-cholesterol uptake in BV2 microglial cells upon ethanol treatment (50 or 100 mM). (**B**) Ethanol effect on ^3^H-cholesterol trafficking to MAM is assayed via its esterification by the MAM-resident enzyme ACAT1 (**C**) Neutral SMase activity in the ethanol-treated (50 and 100 mM) and untreated BV2 cells. Data represent mean ± SEM, *n* = 5 independent experiments. * *p* < 0.05 compared to their respective control counterparts according to the two-way ANOVA followed by Bonferroni’s post hoc test.

**Figure 4 ijms-22-08438-f004:**
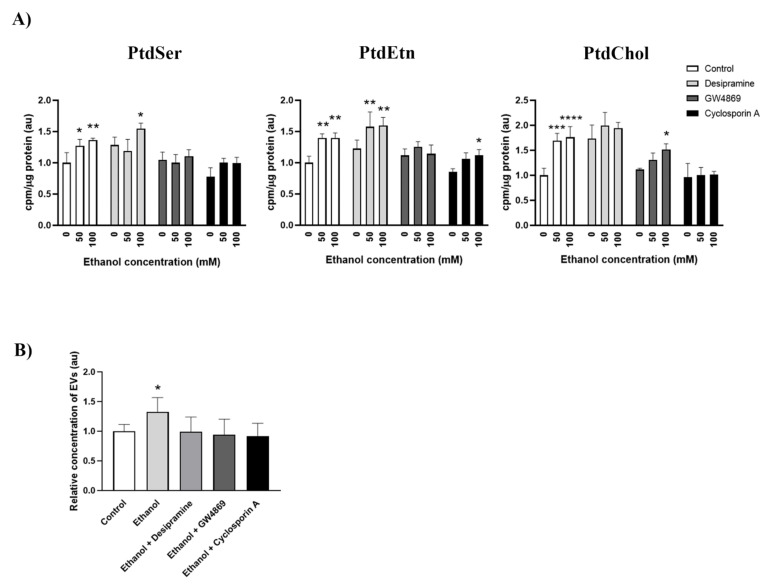
Inhibition of MAM or neutral SMases abrogates ethanol-induced PLT activity and EV secretion in microglia. (**A**) Analysis of the effect of inhibitors for SMases (desipramine, GW4869) or MAM (cyclosporin A) on ethanol-induced phospholipid transfer activity. Cells were preincubated for 2 h with desipramine, GW4869 or cyclosporin A, and then treated with ^3^H-serine along with ethanol (50 or 100 mM) for 4 h. The incorporation of ^3^H-serine into PtdSer, PtdEtn or PtdChol for each group is shown. (**B**) BV2 cells were treated with SMase or MAM inhibitors along with ethanol, as described in Materials and Methods (Section 4.5), and the concentration of EVs with a size between 100–200 nm was quantified via Nanosight for each experimental condition. Graph bars represent fold change over the control. Data represent mean ± SEM, *n* = 5 independent experiments. * *p* < 0.05, ** *p* < 0.01, *** *p* < 0.001 and **** *p* < 0.0001 compared to their respective control group, according to two-way ANOVA (**A**) and Kruskal–Wallis analysis (**B**) with the Bonferroni post hoc test.

## Data Availability

Not applicable.

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
