# Peer review of "Ethanol Induces Extracellular Vesicle Secretion by Altering Lipid Metabolism through the Mitochondria-Associated ER Membranes and Sphingomyelinases"

_ijms, 2021, doi:10.3390/ijms22168438_

Round 1

Reviewer 1 Report

In this study, Ibáñez and collaborators demonstrate that ethanol promotes the secretion of exosomes in BV2 microglial cells by altering the lipid metabolism through the modulation of SMases and MAMs. Despite the scientific relevance of the present study some points need to be addressed:

  1. The approval number, gender and number of animals should be specified in the Material and Methods section. The description of the statistical analysis should also be added;
  2. Figure 1A - a loading control (e.g. ß-actin) should be added;
  3. Figure 1E - poor quality of the representative image from CD81;
  4. What is the mechanism by which cyclosporin A inhibits MAMs?
  5. What happens to mitofusin-2 in these experimental conditions?

Author Response

REVIEWER 1:

 The approval number, gender and number of animals should be specified in the Material and Methods section. The description of the statistical analysis should also be added;

Thank you for your comment. Following your suggestions, we are now included in M&M (pp. 10 and 12) the approval number, gender, the number of animals and the statistical analysis.

Figure 1A - a loading control (e.g. ß-actin) should be added;

Figure 1E - poor quality of the representative image from CD81;

Thank you very much for calling our attention. Following your indications, in Figure 1A it is now included the GAPDH used as a loading control of Western blot, and in the Figure 1E, the image of CD81 has been changed for another one with more quality resolution.

What is the mechanism by which cyclosporin A inhibits MAMs?

Thank you very much for drawing our attention. We have now explained in more detail the mechanism by which cyclosporin A is able to inhibit MAMs (pp. 6-7). In fact, cyclosporin A is a pharmacological inhibitor of Cyclophilin D (CypD), which disrupts the interaction between CypD and IP3R, and concomitantly, the interaction between mitochondria and the ER at the MAM interface.

What happens to mitofusin-2 in these experimental conditions?

Mitofusin-2 knock-out mouse embryonic fibroblasts show a downregulation of phospholipid transfer activity when compared to the WT counterparts (Area-Gomez et al., 2012). However, the effect on ethanol on MFN2-KO models has not been studied yet. Following the reviewer’s suggestion, we analyzed the effect of ethanol (50 and 100 mM) on phospholipid transfer and activity in MFN2-KO cells. We corroborated previous results, demonstrating that unstimulated MFN2-KO cells showed a decrease in phospholipid transfer activity when compared to the unstimulated MFN2-WT group. As expected, ethanol was capable to modulate MAM activity in MFN2-WT cells, as shown in the increased incorporation of 3H-Ser into PS and PC. Interestingly, MFN2-KO cells failed to show these changes, suggesting that the ethanol-induced upregulation of MAM activities depends on a functional ER-mitochondria connection. We cannot fully address why the level of radioactive Ser incorporated into PE showed no changes. A possible explanation could be the ethanol-induced increased conversion of PE into PC. Further understanding of this could help understand the effect of MFN2 and ethanol on mitochondrial membranes but it is out of the scope of this paper.

Reviewer 2 Report

The authors present an interesting and well-focused article. The article has potential. However, there are delicate points that need to be clarified and improved.
-The title must be reevaluated by the authors, the authors must give a more precise title. This article is more of a proof of concept.
-The introduction to the state of the art is well oriented. However, it should be expanded with data that support the spirit of the study and its possible applicability. Please, authors should use more current references.
-The results should be presented in a more orderly manner. The authors must give suggestive and attractive titles that arouse the interest of the reader at all times. The authors should review this point.
-The results and data must be presented with the IQR in all cases, since their results do not follow a normal distribution. Authors should give all p-values.
-Authors must show error bars in all figures.
-The reviewer would like to see the autoradiographs in supplementary material of the WB.
-The authors should make their figures more self-explanatory. The authors should expand the information given in their figure legends.
-Figure 1.B should show a panoramic image and then go to detail. This image is confusing.
-Figure 1.C should show a panoramic image and then another at higher magnifications. In the current format it is confusing and does not support the authors' conclusions.
-The authors make a very brief discussion. This point should be reconsidered and given a more translational approach.
-Figure 4 is confusing and of poor quality. This figure should be better explained, because in the current format it is misleading.
-Why do the authors only do ERK and p-ERk? This point is highly debatable. The authors take an overly conservative approach to the very categorical conclusions they make. Authors should include route points higher and lower.
-The authors do an ANOVA. This is absolutely debatable. The authors must justify the limited sample size they have and such permissive tests are not adequate.
-The authors must provide in supplementary material all the images of the autoradiographs of the WBs and the images without fluorescence composition of figure 1.B. This must be mandatory.
-The limitations of the study should be included, and the possible importance and translation of these results should be highlighted.
-The material and methods is too rare. Authors should expand on the information provided and give current and appropriate citations.
-English is very deficient and must be improved by an expert.
-Finally, authors should consider improving their abstract and their keywords. It is too simple and fails to attract the reader's attention. I would suggest to the authors to include a graphic summary to have more potential.

Author Response

REVIEWER 2:

 - The title must be reevaluated by the authors, the authors must give a more precise title. This article is more of a proof of concept.

Following the reviewer’s comment, the title of the manuscript has been changed by “Ethanol induces extracellular vesicle secretion by altering the lipid metabolism through the mitochondria-associated membrane and sphingomyelinase activation”.

- The introduction to the state of the art is well oriented. However, it should be expanded with data that support the spirit of the study and its possible applicability. Please, authors should use more current references.

As suggested by the reviewer, the Introduction has been expanded and current references has also been included.

- The results should be presented in a more orderly manner. The authors must give suggestive and attractive titles that arouse the interest of the reader at all times. The authors should review this point.

Following the reviewer’s suggestions, the results have been reorganized and the titles of the different sections of the Results have been changed for more attractive ones.

- The results and data must be presented with the IQR in all cases, since their results do not follow a normal distribution. Authors should give all p-values.

-Authors must show error bars in all figures.

As suggested, the data in the results section are now presented with IQR and p-values. In addition, normality tests were also conducted in order to clarify their distribution. In the new version of the manuscript, all figures show error bars.

- The authors should make their figures more self-explanatory. The authors should expand the information given in their figure legends.

Figure legends are now expanded with more information.

- The reviewer would like to see the autoradiographs in supplementary material of the WB.

- Figure 1.B should show a panoramic image and then go to detail. This image is confusing.

- Figure 1.C should show a panoramic image and then another at higher magnifications. In the current format it is confusing and does not support the authors' conclusions.

As the reviewer suggest, we have now included in Supplementary Material the autoradiographs of the Western blot as well as the panoramic and higher magnifications of the fluorescence and electronic microscopy.

- The authors make a very brief discussion. This point should be reconsidered and given a more translational approach.

- The limitations of the study should be included, and the possible importance and translation of these results should be highlighted.

We agree with the reviewer’s comment, and we have included in the Discussion section a more translational approach of the manuscript’s results and study’s limitations (p. 9).

- Figure 4 is confusing and of poor quality. This figure should be better explained, because in the current format it is misleading.

As the reviewer suggests, the figures have been reorganized in the Results section and the Figure 4 is now included in Supplementary material.

- Why do the authors only do ERK and p-ERk? This point is highly debatable. The authors take an overly conservative approach to the very categorical conclusions they make. Authors should include route points higher and lower.

As suggested, we have now included other inflammatory associated signaling pathway, as ERK and p38, as well as the downstream transcription factor, NF-kB.

- The authors do an ANOVA. This is absolutely debatable. The authors must justify the limited sample size they have and such permissive tests are not adequate.

We have now reviewed the statistics of all the data presented, and we have used normality tests to corroborate all data follow a Gaussian distribution. However, when data do not follow a Gaussian distribution, t-test have been replaced by the non-parametric Wilcoxon-Mann-Whitney test and one-ANOVA by Kruskal-Wallis.

- The authors must provide in supplementary material all the images of the autoradiographs of the WBs and the images without fluorescence composition of figure 1.B. This must be mandatory.

All autoradiographs of the Western blot as well as the fluorescence composition of the images of Figure 1B have been included in Supplementary material.

- The material and methods is too rare. Authors should expand on the information provided and give current and appropriate citations.

The different techniques used have been now described in detail in the M&M section and the current references have also been included.

- English is very deficient and must be improved by an expert.

Thank you very much for drawing our attention. The English language has been corrected.

- Finally, authors should consider improving their abstract and their keywords. It is too simple and fails to attract the reader's attention. I would suggest to the authors to include a graphic summary to have more potential.

As indicated, other additional keywords have been included at the end of the Abstract (e.g., secretion, phospholipids and sphingomyelinase) and, as suggested, we have tried to improve the abstract, although the journal recommends a maximum of 200 words.

Round 2

Reviewer 1 Report

Overall, the authors have satisfactorily addressed most of the reviewers' comments. 

Author Response

Thank you very much for the reviewer's comment.

Reviewer 2 Report

The authors have improved the manuscript. But they have overlooked important issues in panoramic images. The reviewer cannot see the supplementary material, it is the same document as the manuscript. In addition, the authors should reference the supplementary material more specifically in the manuscript. Please show the supplementary material to support your conclusions. I invite the authors to make an improvement in English grammar. The manuscript would benefit from a graphic summary to attract the interest of the reader.

Author Response

Thank you very much for the reviewer's comments. As suggested, the Supplementary Material has been referenced more specifically in the manuscript (pp. 4, 10 and 11). We apologize to the reviewer for not being able to see the Supplementary Material, we have re-uploaded the document in Word format.

The English grammar of the manuscript have been revised by Dr. Rishi Agrawal, who was included in the Acknowledgments of the manuscript.

Following the reviewer's suggestion, we have uploaded a graphical abstract.

Round 3

Reviewer 2 Report

The authors have answered correctly and satisfactorily all the doubts and recommendations.